# Variational Approach for Joint Kidney Segmentation and Registration from DCE-MRI Using Fuzzy Clustering with Shape Priors

**DOI:** 10.3390/biomedicines11010006

**Published:** 2022-12-21

**Authors:** Moumen El-Melegy, Rasha Kamel, Mohamed Abou El-Ghar, Norah S. Alghamdi, Ayman El-Baz

**Affiliations:** 1Electrical Engineering Department, Assiut University, Assiut 71515, Egypt; 2Computer Science Department, Assiut University, Assiut 71515, Egypt; 3Radiology Department, Urology and Nephrology Center, Mansoura University, Mansoura 35516, Egypt; 4Department of Computer Sciences, College of Computer and Information Sciences, Princess Nourah bint Abdulrahman University, P.O. Box 84428, Riyadh 11671, Saudi Arabia; 5Bioengineering Department, University of Louisville, Louisville, KY 40292, USA

**Keywords:** chronic kidney disease, kidney segmentation, kidney registration, DCE-MRI, level set, fuzzy c-means, U-Net

## Abstract

The dynamic contrast-enhanced magnetic resonance imaging (DCE-MRI) technique has great potential in the diagnosis, therapy, and follow-up of patients with chronic kidney disease (CKD). Towards that end, precise kidney segmentation from DCE-MRI data becomes a prerequisite processing step. Exploiting the useful information about the kidney’s shape in this step mandates a registration operation beforehand to relate the shape model coordinates to those of the image to be segmented. Imprecise alignment of the shape model induces errors in the segmentation results. In this paper, we propose a new variational formulation to jointly segment and register DCE-MRI kidney images based on fuzzy c-means clustering embedded within a level-set (LSet) method. The image pixels’ fuzzy memberships and the spatial registration parameters are simultaneously updated in each evolution step to direct the LSet contour toward the target kidney. Results on real medical datasets of 45 subjects demonstrate the superior performance of the proposed approach, reporting a Dice similarity coefficient of 0.94 ± 0.03, Intersection-over-Union of 0.89 ± 0.05, and 2.2 ± 2.3 in 95-percentile of Hausdorff distance. Extensive experiments show that our approach outperforms several state-of-the-art LSet-based methods as well as two UNet-based deep neural models trained for the same task in terms of accuracy and consistency.

## 1. Introduction

Chronic kidney disease (CKD) is one of the crucial worldwide public health problems. It is typically defined as heterogeneous disorders causing changes in kidney structure and deterioration in its function [1,2]. Worldwide, the number of people afflicted by this disease is increasing annually, reaching more than 12% of humanity [2]. The patients whose kidneys are totally damaged are diagnosed with end-stage kidney disease. When the patient reaches this stage, she/he has to do either hemodialysis or kidney transplantation [2,3]. Transplantation is indeed more preferable than dialysis as it improves the quality of a patient’s life and returns it close to normal. Early detection and treatment of CKD is important to rescue the patients and avoid achieving end-stage renal failure. The diagnosis of kidney dysfunction is usually done either by estimating the glomerular filtration rate (GFR), which is deemed a key indicator of renal function, or by using a biopsy [2]. Unfortunately, traditional medical tests used for GFR measurement are not effective because failure can only be detected after losing 60% of renal function. Biopsy is known as a gold-standard method. However, it may cause pain, bleeding, and other side consequences to the patient. 

Recently, the medical community has resorted to CAD systems to study human kidney function [4]. DCE-MRI is one of the major non-invasive and safe-modality imaging techniques that are used for accurate renal function assessment as it provides both anatomical and functional kidney information. In DCE-MRI, each patient has a dataset of about 80 kidney scans that are acquired after the patient is injected with a gadoteric acid contrast agent and during the perfusion process into the kidney tissue [5]. Figure 1 shows a set of in-vivo kidney DCE-MRIs of one subject.

CAD systems can efficiently help doctors in making the right diagnosis decision in little time. A typical pipeline for a CAD system is shown in Figure 2. For a specific subject, the kidney is first segmented from each time-point image. As the estimation of perfusion-related parameters mandates that the same pixels are evaluated between the time-point images, a nonrigid alignment operation is carried out. This operation provides a pixel-to-pixel match of the time series images and corrects for patient’s motion or breathing during image acquisition. After that, the renal cortex is extracted from all segmented kidneys, from which, perfusion physiological indexes are estimated. Finally, the diagnosis of kidney status is made.

Obviously, kidney segmentation thus becomes a crucial and prerequisite step for the determination of the kidney status. However, achieving accurate kidney segmentation is very challenging due to the patient’s motion, contrast variation, and low spatial resolution of acquired images.

This paper is organized as follows: First, Section 2 briefly reviews several DCE-MRI kidney segmentation methods developed in the literature. In addition, it discusses the research gap and presents the paper contributions. Then, in Section 3, we describe in detail the ingredients of the proposed approach: the LSet-based energy functional, fuzzy memberships’ computation, statistical SP-model construction, and the affine-based registration process. Next, in Section 4, we present our conducted experiments and the obtained results. Finally, we conclude in Section 5.

## 2. Related Work

Extensive efforts have been done to segment the kidney and its compartments from DCE-MRI data (please refer to a recent survey in [6]). In that regard, the variational level-set (LSet) methods [7,8,9,10,11,12,13,14,15,16,17,18] have been the most successful. These methods employ prior shape information with intensity information [7,8,9,10,11,12,13] and/or a spatial interaction model [14,15,16,17] to guide the contour towards the target kidney in the image. In a different manner, the authors in [18] present an LSet cost functional that simultaneously performs kidney segmentation and motion correction in DCE-MRI data. The main deficiency of this method is that it requires user interaction to manually initialize training masks in the DCE-MRI data before the evolution of the LSet function. Other techniques have also been used for kidney segmentation, such as the discrete wavelet transform [19], k-means clustering with principal component analysis [20], and GrabCut algorithm with a random forest classifier [21]. From a different perspective, Al-Shamasneh et al. [22,23] addressed the low contrast and intensity inhomogeneity problems of DCE-MRI images using fractional calculus. 

Recently, there has been an increasing interest in using convolutional neural networks (CNet) for kidney segmentation tasks [24,25,26,27,28,29,30,31]. The authors in [24] employed transfer learning from a network trained for brain segmentation to a network designated for 3D DCE-MRI kidney segmentation. Haghighi et al. [25] used two cascaded U-Net (UNet) models [26], where the first one performs kidney localization and the second accomplishes the segmentation task. While Bevilacqua et al. [27] present two different CNet-based approaches for accurate MRI kidney segmentation, Brunetti et al. [28] incorporated genetic algorithm with deep learning for the same sake. Later on, Milecki et al. [29] presented a two-step kidney segmentation approach in which they first used thresholding techniques and morphological operators and then developed a 3D unsupervised CNet architecture for extracting kidneys. Isensee et al. [30] achieved the first rank in CHAOS challenge [31] for abdominal organ segmentation, including the left and right kidneys, from MRI data via employing an nnU-Net model.

Deep learning networks generally need an extensive amount of data for proper training, which is unfortunately elusive in the medical field. This presents a major obstacle to CNet-based methods [24,25,26,27,28,29,30,31] towards achieving high segmentation accuracy. LSet-based kidney segmentation methods [7,8,9,10,11,12,13,14,15,16,17,18], on the other hand, have proved their potential in achieving high performance with more accurate segmentation. The majority of these methods depend on using prior information about the kidney’s shape, which calls for a registration operation to align the DCE-MRI images to be segmented to a pre-constructed shape reference model. This registration is typically performed first as a separate pre-processing task before the kidney segmentation task. The main drawback of this is that any errors in this registration step significantly affect segmentation performance. 

In an attempt to alleviate this limitation, inspired by earlier work [32] in the context of brain MRI segmentation, this paper aims to propose—for the first time—a variational approach for joint kidney segmentation and registration based on FCM clustering [33] embedded within an LSet method [34]. The approach constrains LSet evolution by shape prior information and the intensity information represented in the fuzzy memberships. The main contributions of this paper can be summarized as follows: First, we formulate a new energy functional to drive the LSet contour towards the target kidney and to simultaneously align this contour with a pre-constructed kidney’s shape prior model (SP-model). Second, in each evolution step, the fuzzy memberships of the image pixels in both the kidney and background clusters are updated concurrently with updating the spatial transformation parameters aligning the image with the SP-model. That is, in each evolution step, the SP-model is gradually transformed according to the spatial transformation parameters so as to be well-aligned with the target kidney. Third, to ensure the robustness of the proposed approach against contour initialization, we employ smeared-out Heaviside and Dirac delta functions in the LSet method. As such, our approach is able to accurately segment the kidney from the image regardless of where the contour has been initialized. Fourth, the SP-model is built off-line from a number of kidney images from various patients by adopting an efficient Bayesian parameter estimation method [35]. This method allows the SP-model to accommodate the possibility of the existence of kidney pixels in the test image where they were not observed in the images used to build the model.

This new approach was applied to 45 patients’ datasets, and its performance was evaluated using three popular segmentation metrics [36,37]: the Dice similarity coefficient (DC), Intersection-over-Union (IoU), and 95-percentile of Hausdorff distance (95HD). To verify the competency of the proposed approach, we conducted comparative experiments with several recent LSet-based methods. We furthermore compared our approach’s performance to that of two deep neural network models designed for this very same task: a UNet model [26] and one of its later amendments named the BCD-UNet model [38]. Both networks were trained from scratch on our DCE-MRI data enlarged by the KiTS19 challenge dataset [39]. Our extensive experiments and comparisons confirmed the high accuracy, consistency, and robustness of the proposed approach against all the other methods. 

## 3. Methods

In this section, we introduce the mathematical formulation of the proposed joint kidney segmentation and registration approach using the LSet method and FCM clustering.

### 3.1. Problem Statement and Notations

For analyzing kidney function, the kidney is to be segmented from each image in the sequence. Let ℕ be the total number of time-point images in the dataset and It={ It(x,y), (x,y)∈Ω , t=1,..,ℕ} be a DCE-MRI grayscale kidney image captured at time *t* that requires accurate segmentation, where It(x,y) is the intensity value of the pixel at location (x,y) in the image domain Ω. Each pixel (x,y) in the image is to be labelled as kidney (K) or background (ℬ).

### 3.2. Proposed Variational Approach

The LSet method [34] is an efficient and high-performance technique that has been extensively used in biomedical image applications, such as image segmentation and registration. It depends on evolving a contour within the image domain according to a predefined energy functional. For evaluating the kidney function of a specific subject, the kidney needs to be segmented from each time-point image in the subject’s sequence. Using an LSet-based method, this can be achieved via performing the following main processing steps. First, an SP-model of the kidney is trained offline from a set of DCE-MRI images. Then, the input image is co-aligned to the constructed shape model adopting a 2D affine transformation [40]. Finally, the LSet contour is iteratively evolved in the image domain and stopped when it captures the kidney’s boundary. Indeed, incorrect alignment between the input image and the SP-model leads to segmentation errors and consequently causes a drop in the method’s performance. Furthermore, performing accurate manual registration between the SP-model and each time-point image is time-consuming and error-prone.

We here propose a new variational approach for the simultaneous tasks of kidney segmentation and registration. The flowchart of the proposed approach is illustrated in Figure 3. As can be seen from the figure, there are two phases in this approach. In the first offline phase, a Bayesian parameter estimation method is adopted to construct an SP-model from a set of co-aligned DCE-MRIs. This phase is performed once for all subjects. It is described in detail in Section 3.5. 

In the second phase shown in Figure 3, the kidney is segmented from each time-point image in a given subject’s sequence. Several initialization steps (Steps 3–6) are first carried out. The LSet contour is initialized near the image borders or even randomly. Initial values for the spatial transformation parameters are also specified. Then, starting out with initial centroid values of the kidney and background clusters, the pixel-wise kidney and background fuzzy memberships are initially computed. Afterwards, a number of steps (Steps 7–10) are iterated until convergence of the energy functional described in Section 3.3. The LSet contour is evolved one step in the direction minimizing this energy functional taking into account the already-constructed SP-model and the current values of the fuzzy memberships of the image pixels inside and outside the contour. Then, the clusters’ centroid values and the image pixels’ fuzzy memberships are updated as detailed in Section 3.4. Moreover, the spatial transformation parameters are updated as explained in Section 3.6. Once the iterative part of the approach is terminated, the final LSet contour designates the output segmented kidney. This entire procedure is repeated for all DCE-MRI images in the input sequence. 

The following subsections detail all the steps of the approach in Figure 3.

### 3.3. Proposed Energy Functional

In this work, we propose a new energy functional consisting of three energy terms: a regularizing length term, a term based on FCM-clustering (FCMC), and a registration term. Let $\Omega_{C}$ be an LSet contour dividing the image domain Ω into two separate regions, kidney $\Omega_{K}$ and background $\Omega_{ℬ}$. The contour is represented by an LSet function ϕ which is positive for each pixel (x,y) in the kidney region, ϕ(x,y)>0, negative in the background region, ϕ(x,y)<0, and zero on the contour ϕ(x,y)=0, as illustrated in Figure 4. 

The proposed joint energy functional is defined as:(1)$$E\left(\phi ,\;A,T\right)=\lambda_{1}\;ℒ\left(\phi \right)+\lambda_{2}\;E_{ℱCℳ}\left(\phi ,\;A,T\right)++\lambda_{3}\;E_{ℛEG}\left(\phi ,\;A,T\right)$$
where all $\lambda_{i}$ are positive weighting coefficients. A and T are affine transformation parameters that are elaborated in Section 3.6. The length term
ℒ(ϕ)
constrains the Lset contour smoothness and is defined as:(2)$$ℒ\left(\phi \right)=\overset{\;}{\underset{\Omega }{\int }}\delta \phi_{\varepsilon }\;\left|\nabla \phi \left(x,y\right)\right|\;dx\;dy\;\;$$
where $\delta \phi_{\varepsilon }=\delta_{\varepsilon }\left(\phi \left(x,y\right)\right)$ is the smoothed Dirac function which equals the derivative of the smeared-out Heaviside function $V\phi_{\varepsilon }=V_{\varepsilon }\left(\phi \left(x,y\right)\right)$, both given by
(3)$$V\phi_{\varepsilon }=\{\begin{array}{cc} 1 & \phi >\varepsilon \\ \frac{1}{2}+\frac{\phi }{2\varepsilon }+\frac{1}{2\text{π}}\sin\left(\frac{\text{π}\phi }{\varepsilon }\right) & -\varepsilon \leq \phi \leq \varepsilon \\ 0 & \phi <-\varepsilon \end{array}\;\delta \phi_{\varepsilon }=\{\begin{array}{cc} 0 & \left|\phi \right|>\varepsilon \\ \frac{1}{2\varepsilon }+\frac{1}{2\varepsilon }\cos\left(\frac{\pi \phi }{\varepsilon }\right) & \left|\phi \right|\leq \varepsilon \end{array}$$
where ε stands for the width of numerical smearing. The energy functional $E_{ℱCℳ}\left(\phi ,\;A,T\right)$ in (1) is basically computed from the input image and has the principal role in the evolution process, and is defined as
(4)$$E_{ℱCℳ}\left(\phi ,\;A,T\right)=\overset{\;}{\underset{\Omega }{\int }}V\phi_{\varepsilon }\;{𝓂}_{ℬ}\left(x,y\right)\;\;P_{ℬ}\left(\hat{x},\hat{y}\right)\;dx\;dy+\overset{\;}{\underset{\Omega }{\int }}\left(1-V\phi_{\varepsilon }\right)\;{𝓂}_{K}\left(x,y\right)\;\;P_{K}\left(\hat{x},\hat{y}\right)\;dx\;dy\;\;$$
where ${𝓂}_{L}\left(x,y\right)$ represents the membership value of the pixel (x,y) in the L-th cluster, L∈{K,ℬ}. $P_{L}\left(\hat{x},\hat{y}\right)$ represents the probability of the pixel $\left(\hat{x},\hat{y}\right)$ in the SP-model corresponding to the image point (x,y) being kidney (L=K) or background (L=ℬ). How the two pixels (x,y) and $\left(\hat{x},\hat{y}\right)$ are related is explained in Section 3.6. Lastly, $E_{ℛEG}\left(\phi ,\;A,T\right)$ in (1) is defined as the squared difference between the Heaviside of the LSet function of the current evolving contour and the Heaviside of the SP-model
(5)$$E_{ℛEG}\left(\phi ,\;A,T\right)=\overset{\;}{\underset{\Omega }{\int }}\;{\left[V\phi_{\varepsilon }-V_{\varepsilon }\left(\phi_{P_{K}}\left(\hat{x},\hat{y}\right)\right)\right]}^{2}\;dx\;dy\;$$

Thus, the basic role of $E_{ℛEG}\left(\phi ,\;A,T\right)$ is to stop the LSet contour when it reaches a shape similar to the kidney shape in the SP-model. By minimizing the energy functional (1) with respect to the LSet function ϕ, we can obtain the following from the calculus of variations: (6)$$\frac{\partial \phi }{\partial t}=\lambda_{1}\delta \phi_{\varepsilon }\;\mathrm{div}\left(\frac{\nabla \phi }{\left|\nabla \phi \right|}\right)+\;\lambda_{2}\;\delta \phi_{\varepsilon }\left[\;{𝓂}_{K}\left(x,y\right)\;\;P_{K}\left(\hat{x},\hat{y}\right)-{𝓂}_{ℬ}\left(x,y\right)\;\;P_{ℬ}\left(\hat{x},\hat{y}\right)\;\right]-\;2\lambda_{3}\;\delta \phi_{\varepsilon }\;\left[V\phi_{\varepsilon }-V_{\varepsilon }\left(\phi_{P_{K}}\left(\hat{x},\hat{y}\right)\right)\;\right]$$

Finally, the LSet contour iteratively evolves towards the kidney boundary in the image according to
(7)$$\phi_{n+1}\left(x,y\right)=\phi_{n}\left(x,y\right)+\gamma_{1}\;\frac{\partial \phi_{n}\left(x,y\right)}{\partial t}$$
where n is a time step and $\gamma_{1}$ is a positive step size. It should be noted that exploiting the smooth Heaviside and Dirac-delta functions enables us to achieve a global minimizer for the functional in (1) even if the initial LSet contour is totally out of the kidney position [34]. 

### 3.4. FCMC Membership Function

The FCMC algorithm [33] is an important and widely used clustering method in different image segmentation applications. It is used to segment an input image into a number of clusters such that pixels in the same cluster have more similarity than those in other clusters. More specifically, the objective of the FCMC algorithm is to segment a DCE-MRI image $I_{t}$ into kidney and background clusters (see Figure 5) via minimizing the subsequent objective function [41]
(8)$$J=\sum_{\left(x,y\right)\;\in \;\Omega \;}\;\sum_{L}\;{𝓂}_{L}^{2}\left(x,y\right)\;||\;I_{t}\left(x,y\right)-C_{L}||{}^{2},$$
where ||.|| denotes the Euclidean distance between the pixel’s intensity and the centroid of kidney cluster $C_{K}$ or background cluster $C_{ℬ}$. ${𝓂}_{L}\left(x,y\right)\in \left[0,1\right]$ is the membership function that represents the degree of pixel belongingness to kidney (L=K) or background (L=ℬ) clusters such that $\sum_{L}{𝓂}_{L}\left(x,y\right)=1$. The membership values of pixels to a specific cluster represent the pixel-wise probabilities belonging to this cluster and depend on the distances between the intensity of pixels and the centroid value. The closer the pixels’ intensities are to the centroid value of a certain cluster, the higher their membership values to this cluster, and vice versa.

In our approach, the computation of centroid values and fuzzy memberships is held in conjunction with the LSet contour evolution. We first initialize kidney and background clusters’ centroids as the average of intensity values for pixels in and out the LSet contour, respectively. Then, the LSet contour starts its evolution, and during this time, the clusters’ centroids and fuzzy memberships of each pixel (x,y) are iteratively computed and updated via
(9)$${𝓂}_{L}\left(x,y\right)=\frac{||\;I_{t}\left(x,y\right)-C_{L}||{}^{-2}}{||\;I_{t}\left(x,y\right)-C_{K}||{}^{-2}+||I_{t}\left(x,y\right)-C_{ℬ}||{}^{-2}}$$
(10)$$C_{L}=\frac{\;\sum_{\left(x,y\right)\in \Omega }{ℛ}_{L}\left(\phi \left(x,y\right)\right)\;\;I_{t}\left(x,y\right)\;{𝓂}_{L}^{2}\left(x,y\right)\;\;}{\sum_{\left(x,y\right)\in \Omega }\;{ℛ}_{L}\left(\phi \left(x,y\right)\right)\;\;{𝓂}_{L}^{2}\left(x,y\right)\;\;}$$
where$\;{ℛ}_{L}\left(\phi \left(x,y\right)\right)$ denotes the smeared-out Heaviside function $V\phi_{\varepsilon }$ for (L=K) and $\left(1-V\phi_{\varepsilon }\right)$ for (L=ℬ). 

It is important to note here that, as depicted in Figure 5, depending only on fuzzy memberships in the kidney segmentation problem is often impractical. This is because it cannot separate the two classes successfully based only on the intensity information. Thus, we incorporate shape prior information with fuzzy memberships in our proposed approach. 

### 3.5. Statistical Shape Prior Model

One of the main advantages of the LSet method is its ability of incorporating prior knowledge about the organ shape. Human kidneys tend to have common shapes. Therefore, using prior information about the kidney shape can significantly enhance the segmentation accuracy. The SP-model is generally constructed adopting the following strategy: First, a number N of DCE-MRIs are selected from several subjects. Then, one of these images is chosen as a reference image to which all images are mutually co-aligned using maximization of mutual information [40]. Eventually, these registered images are segmented by a medical expert, and the SP-model is trained from the obtained ground-truth segmentations. Some earlier studies (e.g., [10,11,12]) adopted a simple, first-order method for computing the pixel-wise probability of the SP-model. However, this method tends not to be accurate, especially if a pixel is labelled as kidney or background in the whole training cohort. In such cases, the pixel is assigned an exact value of 1 for the observed label and 0 for the unobserved label, which is often unreasonable.

In this study, we overcome this problem via employing the Bayesian parameter estimation method [35] in the construction of the SP-model in the following manner: The kidney/background probability of pixel $\left(\hat{x},\hat{y}\right)$, whenever the kidney and background labels are both noticed, is computed according to [35,42]
(11)$$P_{L}\left(\hat{x},\hat{y}\right)=\left[\frac{N_{L}\left(\hat{x},\hat{y}\right)+\beta }{\;\;N+\beta \;O\left(\hat{x},\hat{y}\right)\;\;}\right]\left[\frac{N}{\;\;N+\ell -O\left(\hat{x},\hat{y}\right)\;\;}\right]$$
where ℓ=2 represents the number of all potential labels (i.e., kidney and background), $P_{L}\left(\hat{x},\hat{y}\right)\in \left[0,\;1\right]$, and $\overset{\;}{\underset{L}{\sum }}P_{L}\left(\hat{x},\hat{y}\right)=1$. In this case, the count of observed labels$\;O\left(\hat{x},\hat{y}\right)$ equals 2 since the pixel is labeled as kidney in a set of DCE-MRIs, but meanwhile, it is observed as background in another set. $N_{L}\left(\hat{x},\hat{y}\right)$ denotes how often the label L is noticed and β is a positive additive weight. In other cases, $O\left(\hat{x},\hat{y}\right)$ equals 1 specifically when the pixel $\left(\hat{x},\hat{y}\right)$ is labeled as either kidney or background throughout the training set. Here, the observed label’s probability is calculated from (11), and the likelihood of an unobserved label is obtained from:(12)$$P_{L}\left(\hat{x},\hat{y}\right)=\left[\frac{1}{\;\;\ell -O\left(\hat{x},\hat{y}\right)}\right]\left[1-\frac{N}{N+\ell -O\left(\hat{x},\hat{y}\right)}\right]$$

According to the above steps, a more discriminative shape model is constructed, as shown in Figure 6. It is worth noting that the Bayesian parameter estimation method accounts for the likelihoods of unobserved labels and generates smooth probabilities. Therefore, it can handle the variation between the shape of the input kidney and the kidneys used in the SP-model construction better than the first-order method used in some other approaches (e.g., [10,11,12]).

### 3.6. Affine-Based Registration for the Shape Prior Model

Renal function assessment from DCE-MRI mainly requires accurate segmentation of kidneys from surrounding structures. This in turn relies on good registration between the prior shape and the input image. We thus seek to find the optimal pixel-wise affine transformation matrix that relates the two. Affine transformation allows for translation, scaling, and shearing without changing the basic geometry of the kidney shape. By assuming 2D affine transformation, each pixel (x,y) in the image space is transformed into another pixel $\left(\hat{x},\hat{y}\right)$ in the reference space of the SP-model such that:(13)$$\hat{X}=A\;X+T$$
where
(14)$$\;\;\;X=\left[\begin{array}{c} x\\ y \end{array}\right]\;\;\;,\;\;\;\hat{X}=\left[\begin{array}{c} \hat{x}\\ \hat{y} \end{array}\right]\;,\;\;A=R\;H\;S\;$$
with R, H, S, and T being rotation, shearing, scaling, and translation matrices, defined as:(15)$$R=\left[\begin{array}{cc} \cos\theta & -\sin\theta \\ \sin\theta & \;\cos\theta \end{array}\right]\;,\;\;H=\left[\begin{array}{cc} 1 & h_{x}\\ h_{y} & 1 \end{array}\right]\;,\;\;S=\left[\begin{array}{cc} s_{x} & 0\\ 0 & s_{y} \end{array}\right]\;,\;\;T=\left[\begin{array}{c} t_{x}\\ t_{y} \end{array}\right]$$
where $s_{x/y}$ and $h_{x/y}$ specify scaling and shearing in x/y directions, whereas $t_{x/y}$ identifies translations along the two orthogonal axes. Minimizing the energy functional (1) with respect to S, H, T, and θ for fixed ϕ yields
(16)$$\frac{\partial E}{\partial S}=\lambda_{2}\overset{\;}{\underset{\Omega }{\int }}V\phi_{\varepsilon }\left(X\right)\;{𝓂}_{ℬ}\left(X\right)\;\left[\nabla P_{ℬ}\left(\hat{X}\right)\;\frac{\partial \hat{X}}{\partial S}\right]\;dX+\lambda_{2}\overset{\;}{\underset{\Omega }{\int }}\left[1-V\phi_{\varepsilon }\left(X\right)\right]\;\;{𝓂}_{K}\left(X\right)\;\;\left[\nabla P_{K}\left(\hat{X}\right)\;\frac{\partial \hat{X}}{\partial S}\right]\;dX\;-2\;\lambda_{3}\;\overset{\;}{\underset{\Omega }{\int }}\delta_{\varepsilon }\left(\phi_{P_{K}}\left(\hat{X}\right)\right)\left[\;V\phi_{\varepsilon }\left(X\right)-V_{\varepsilon }\left(\phi_{P_{K}}\left(\hat{X}\right)\right)\;\right]\;\left[\nabla \phi_{P_{K}}\left(\hat{X}\right)\;\frac{\partial \hat{X}}{\partial S}\right]\;dX\;\frac{\partial E}{\partial H}=\lambda_{2}\overset{\;}{\underset{\Omega }{\int }}V\phi_{\varepsilon }\left(X\right)\;{𝓂}_{ℬ}\left(X\right)\;\left[\nabla P_{ℬ}\left(\hat{X}\right)\;\frac{\partial \hat{X}}{\partial H}\right]\;dX+\lambda_{2}\overset{\;}{\underset{\Omega }{\int }}\left[1-V\phi_{\varepsilon }\left(X\right)\right]\;\;{𝓂}_{K}\left(X\right)\;\left[\nabla P_{K}\left(\hat{X}\right)\;\frac{\partial \hat{X}}{\partial H}\right]\;dX\;-2\;\lambda_{3}\;\overset{\;}{\underset{\Omega }{\int }}\delta_{\varepsilon }\left(\phi_{P_{K}}\left(\hat{X}\right)\right)\left[\;V\phi_{\varepsilon }\left(X\right)-V_{\varepsilon }\left(\phi_{P_{K}}\left(\hat{X}\right)\right)\;\right]\;\left[\nabla \phi_{P_{K}}\left(\hat{X}\right)\;\frac{\partial \hat{X}}{\partial H}\right]\;dX\;\frac{\partial E}{\partial T}=\lambda_{2}\overset{\;}{\underset{\Omega }{\int }}V\phi_{\varepsilon }\left(X\right)\;{𝓂}_{ℬ}\left(X\right)\;\left[\nabla P_{ℬ}\left(\hat{X}\right)\;\frac{\partial \hat{X}}{\partial T}\right]\;dX+\lambda_{2}\overset{\;}{\underset{\Omega }{\int }}\left[1-V\phi_{\varepsilon }\left(X\right)\right]\;\;{𝓂}_{K}\left(X\right)\;\left[\nabla P_{K}\left(\hat{X}\right)\;\frac{\partial \hat{X}}{\partial T}\right]\;dX\;-2\;\lambda_{3}\;\overset{\;}{\underset{\Omega }{\int }}\delta_{\varepsilon }\left(\phi_{P_{K}}\left(\hat{X}\right)\right)\left[\;V\phi_{\varepsilon }\left(X\right)-V_{\varepsilon }\left(\phi_{P_{K}}\left(\hat{X}\right)\right)\;\right]\;\left[\nabla \phi_{P_{K}}\left(\hat{X}\right)\;\frac{\partial \hat{X}}{\partial T}\right]\;dX\;\frac{\partial E}{\partial \theta }=\lambda_{2}\overset{\;}{\underset{\Omega }{\int }}V\phi_{\varepsilon }\left(X\right)\;{𝓂}_{ℬ}\left(X\right)\;\left[\nabla P_{ℬ}\left(\hat{X}\right)\;\frac{\partial \hat{X}}{\partial \theta }\right]\;dX+\lambda_{2}\overset{\;}{\underset{\Omega }{\int }}\left[1-V\phi_{\varepsilon }\left(X\right)\right]\;\;{𝓂}_{K}\left(X\right)\;\left[\nabla P_{K}\left(\hat{X}\right)\;\frac{\partial \hat{X}}{\partial \theta }\right]\;dX\;-2\;\lambda_{3}\;\overset{\;}{\underset{\Omega }{\int }}\delta_{\varepsilon }\left(\phi_{P_{K}}\left(\hat{X}\right)\right)\left[\;V\phi_{\varepsilon }\left(X\right)-V_{\varepsilon }\left(\phi_{P_{K}}\left(\hat{X}\right)\right)\;\right]\;\left[\nabla \phi_{P_{K}}\left(\hat{X}\right)\;\frac{\partial \hat{X}}{\partial \theta }\right]\;dX$$
where ∇(·) is the gradient through all directions. $\frac{\partial \hat{X}}{\partial S}\;,\;\frac{\partial \hat{X}}{\partial H}\;,$ $\frac{\partial \hat{X}}{\partial T}$ , and $\frac{\partial \hat{X}}{\partial \theta }$ are the differentiation of the transformed coordinates $\hat{X}$ with respect to registration parameters S, H**,**
T, and θ such that:(17)$$\;\frac{\partial \hat{X}}{\partial S}={\left[\begin{array}{cc} \frac{\partial \hat{X}}{\partial s_{x}} & \frac{\partial \hat{X}}{\partial s_{y}} \end{array}\right]}^{T}\;\;\;,\;\;\;\;\;\;\frac{\partial \hat{X}}{\partial H}={\left[\begin{array}{cc} \frac{\partial \hat{X}}{\partial h_{x}} & \frac{\partial \hat{X}}{\partial h_{y}} \end{array}\right]}^{T}\;\;\;\;\;\;,\;\;\frac{\partial \hat{X}}{\partial T}={\left[\begin{array}{cc} \frac{\partial \hat{X}}{\partial t_{x}} & \frac{\partial \hat{X}}{\partial t_{y}} \end{array}\right]}^{T}$$

It should be noted that, in our implementation, (16) is transformed to its discretized form as follows: $$\frac{\partial E}{\partial S}=\lambda_{2}\overset{N}{\underset{i=1}{\sum }}V\phi_{\varepsilon }\left(X_{i}\right)\;\;{𝓂}_{ℬ}\left(X_{i}\right)\;\;\nabla P_{ℬ}\left({\hat{X}}_{i}\right)\;\frac{\partial \hat{X}}{\partial S}+\lambda_{2}\overset{N}{\underset{i=1}{\sum }}\left[1-V\phi_{\varepsilon }\left(X_{i}\right)\right]\;\;{𝓂}_{K}\left(X_{i}\right)\;\;\nabla P_{K}\left({\hat{X}}_{i}\right)\;\frac{\partial \hat{X}}{\partial S}\;\;\;\;-2\;\lambda_{3}\overset{N}{\underset{i=1}{\sum }}\delta_{\varepsilon }\left(\phi_{P_{K}}\left({\hat{X}}_{i}\right)\right)\;\;\left[\;V\phi_{\varepsilon }\left(X_{i}\right)-V_{\varepsilon }\left(\phi_{P_{K}}\left({\hat{X}}_{i}\right)\right)\;\right]\;\nabla \phi_{P_{K}}\left({\hat{X}}_{i}\right)\;\frac{\partial \hat{X}}{\partial S}\;$$
(18)$$\frac{\partial E}{\partial H}=\lambda_{2}\overset{N}{\underset{i=1}{\sum }}V\phi_{\varepsilon }\left(X_{i}\right)\;\;{𝓂}_{ℬ}\left(X_{i}\right)\;\;\nabla P_{ℬ}\left({\hat{X}}_{i}\right)\;\frac{\partial \hat{X}}{\partial H}+\lambda_{2}\overset{N}{\underset{i=1}{\sum }}\left[1-V\phi_{\varepsilon }\left(X_{i}\right)\right]\;\;{𝓂}_{K}\left(X_{i}\right)\;\;\nabla P_{K}\left({\hat{X}}_{i}\right)\;\frac{\partial \hat{X}}{\partial H}\;\;\;-2\;\lambda_{3}\overset{N}{\underset{i=1}{\sum }}\delta_{\varepsilon }\left(\phi_{P_{K}}\left({\hat{X}}_{i}\right)\right)\;\;\left[\;V\phi_{\varepsilon }\left(X_{i}\right)-V_{\varepsilon }\left(\phi_{P_{K}}\left({\hat{X}}_{i}\right)\right)\;\right]\;\nabla \phi_{P_{K}}\left({\hat{X}}_{i}\right)\;\frac{\partial \hat{X}}{\partial H}\;\;\frac{\partial E}{\partial T}=\lambda_{2}\overset{N}{\underset{i=1}{\sum }}V\phi_{\varepsilon }\left(X_{i}\right)\;\;{𝓂}_{ℬ}\left(X_{i}\right)\;\;\nabla P_{ℬ}\left({\hat{X}}_{i}\right)\;\frac{\partial \hat{X}}{\partial T}+\lambda_{2}\overset{N}{\underset{i=1}{\sum }}\left[1-V\phi_{\varepsilon }\left(X_{i}\right)\right]\;\;{𝓂}_{K}\left(X_{i}\right)\;\;\nabla P_{K}\left({\hat{X}}_{i}\right)\;\frac{\partial \hat{X}}{\partial T}\;\;\;-2\;\lambda_{3}\overset{N}{\underset{i=1}{\sum }}\delta_{\varepsilon }\left(\phi_{P_{K}}\left({\hat{X}}_{i}\right)\right)\;\;\left[\;V\phi_{\varepsilon }\left(X_{i}\right)-V_{\varepsilon }\left(\phi_{P_{K}}\left({\hat{X}}_{i}\right)\right)\;\right]\;\nabla \phi_{P_{K}}\left({\hat{X}}_{i}\right)\;\frac{\partial \hat{X}}{\partial T}\;\;\frac{\partial E}{\partial \theta }=\lambda_{2}\overset{N}{\underset{i=1}{\sum }}V\phi_{\varepsilon }\left(X_{i}\right)\;\;{𝓂}_{ℬ}\left(X_{i}\right)\;\;\nabla P_{ℬ}\left({\hat{X}}_{i}\right)\;\frac{\partial \hat{X}}{\partial \theta }+\lambda_{2}\overset{N}{\underset{i=1}{\sum }}\left[1-V\phi_{\varepsilon }\left(X_{i}\right)\right]\;\;{𝓂}_{K}\left(X_{i}\right)\;\;\nabla P_{K}\left({\hat{X}}_{i}\right)\;\frac{\partial \hat{X}}{\partial \theta }\;\;\;-2\;\lambda_{3}\overset{N}{\underset{i=1}{\sum }}\delta_{\varepsilon }\left(\phi_{P_{K}}\left({\hat{X}}_{i}\right)\right)\;\;\left[\;V\phi_{\varepsilon }\left(X_{i}\right)-V_{\varepsilon }\left(\phi_{P_{K}}\left({\hat{X}}_{i}\right)\right)\;\right]\;\nabla \phi_{P_{K}}\left({\hat{X}}_{i}\right)\;\;\frac{\partial \hat{X}}{\partial \theta }\;$$
where N is the total number of pixels in the image. Eventually, the optimal affine transformation between the SP-model and the input image is sought by iteratively updating S, H,T, and θ in each evolution iteration via
(19)$$S_{n+1}=S_{n}-\gamma_{2}\;\frac{\partial E}{\partial S}\;\;H_{n+1}=H_{n}-\gamma_{3}\;\frac{\partial E}{\partial H}\;T_{n+1}=T_{n}-\gamma_{4}\;\frac{\partial E}{\partial T}\;\theta_{n+1}=\theta_{n}-\gamma_{5}\;\frac{\partial E}{\partial \theta }$$
where $\gamma_{i}$, *i* = 2, …, 5, are positive constants. In our experiments, θ is initially set to 0, H and S are initialized as identity matrices, and T is initially a zero vector.

## 4. Results

This section provides a set of experiments to validate the proposed approach on 45 subjects’ DCE-MRI datasets. To evaluate the proposed approach’s accuracy, we measure the similarity between the obtained segmentation results and the ground-truth (manually segmented) kidneys by computing the mean and standard deviation of the DC, IoU, and 95HD metrics [36,37]. The higher the first two metrics, the better the segmentation, while a lower 95HD value indicates better segmentation. We used the Bayesian parameter estimation method to construct the SP-model from ground-truth kidneys from 30 subjects, one image per each of them. The parameters used for the proposed approach were experimentally determined and were fixed to $\left(\lambda_{1}\;,\;\lambda_{2},\;\lambda_{3},\;\varepsilon ,\;\beta ,\;\gamma_{1},\;\gamma_{2}\;,\;\gamma_{3}\;,\;\gamma_{4}\;,\;\gamma_{5}\right)=$ $\left(6\;,\;6\;,\;0.1,\;1.5,\;1,\;0.8,\;1\times 10^{-14},\;1\times 10^{-10}\;,\;1\times 10^{-10},\;1\times 10^{-9}\right)$ throughout all the conducted experiments without any further tuning.

### 4.1. Data

For this study, we used real datasets of 45 patients who had kidney transplant surgery at Mansoura University Hospital, Egypt. Each dataset contained approximately 80 256 × 256 DICOM scans acquired via patient injection with a Gd-DTPA contrast agent at a speed of 3–4 mL/s and dose of 0.2 mL/kgBW. During this time, repeated kidney scans were captured at 3 sec intervals employing a 1.5T MRI scanner with torso phased-array coils. The acquired scans were manually segmented by an experienced radiologist with more than 10 years of hands-on-experience at the hospital. Figure 1 shows the intensity variation of acquired scans of one patient caused by the agent’s transition into the blood stream.

### 4.2. Comparison with Other LSet-Based Methods

The performance of the proposed approach was first evaluated on the collected DCE-MRIs. For emphasizing the proposed approach’s efficacy, we augmented the test data through applying affine transformations with random rotation angles between −2 and +2 degrees and random shearing in range [0, 12] to all images in each subject’s sequence. Thus, the size of each subject’s dataset was doubled to include the original DCE-MRI images as well as those affine-transformed images. These additional transformed images were intended to make the registration of the images with the reference shape model more challenging for the proposed approach. Figure 7 demonstrates the benefits of using the proposed approach for kidney segmentation. 

Accurate registration between the image in the top row of Figure 7 with the shape reference model led to good segmentation of the final result in (c) with a high Dice similarity coefficient (DC) of 0.96. On the other hand, inaccurate registration in the bottom row propagated errors to the segmentation process resulting in a lower DC of 0.70 for the corresponding final result in (c). Performing registration and segmentation simultaneously produced more accurate results in column (d) for the two images with a DC value of 0.97 in both cases.

The performance of the proposed approach was then verified by comparing it with that of very recent LSet-based methods, namely FCMLS [11], PBPSFL [12], PSFL [13], and FML [17]. It is noteworthy that each of these LSet-based methods [11,12,13,17] employs a prior shape model; thus, for fair comparison, this model was constructed from the same training cohort that we used to build the SP-model in our proposed approach. The initial LSet contour was deliberately located away from the kidney in all conducted experiments. Table 1 reports the obtained results. It reports the three metrics on all the DCE-MRI images as well as on the particular subset of the additional, affine-transformed images. Figure 8 visually confirms the results reported in Table 1.

As observed from Table 1, the proposed approach performed substantially better than the other methods, reporting the highest mean DC and IoU values as well as the lowest mean 95HD. It also exhibited a more consistent performance with lower standard deviation values than the other existing methods. In those methods, the alignment with the shape model was done before the segmentation process was run. As such, their performance manifestly dropped when the input image was not properly aligned to the model. This was more evident from their lower performance on the additional transformed subset. In contrast, our new approach demonstrated its capability to handle such cases and converged to more accurate results. This can be attributed to the simultaneous employment of kidney segmentation and SP-model registration.

As depicted in the first two rows of Figure 8, the proposed approach has shown segmentation performance to comparable that of existing methods when the image was perfectly aligned with the SP-model. However, in the case of inaccurate alignment, the other methods could not segment the kidneys out from the background and generated unsatisfactory segmentation results. Conversely, our approach correctly guided the LSet contour to the target kidney and output precise segmentation results. The comparative results proved that the proposed approach had better ability to rectify the misalignment between the SP-model and the target kidney, producing better segmentations. 

We then carried out a series of experiments to inspect the sensitivity of the proposed approach to the LSet contour initialization. Figure 9 illustrates examples for the output segmentations by our approach when initiated with three different contour locations. Based on the reported DC values, the initial contour location had almost no influence on the proposed approach’s performance. This indeed emphasizes the high-reliability and full-automation of this new approach.

### 4.3. Comparison with UNet-Based Convolutional Neural Networks

Deep neural networks based on the UNet model and its later modifications have become increasingly popular for medical image segmentation tasks. Therefore, we here compared the performance of the proposed approach with that of the basic UNet model [26] and one of its recent variants called the BCD-UNet model [38]. Both networks were trained from scratch using datasets of 18 and 12 subjects for training and validation, respectively, while the remaining subjects’ datasets were used as held-out test data. In order to make the trained networks more robust and to improve their performances, we augment the training and validation data by applying the following to each image: (1) vertical and horizontal flipping, (2) random x-and-y translations, (3) rotation by (±45°, ±90°, 180°) angles, and (4) zero mean Gaussian noise with variance 0.01, 0.02, and 0.05 (after normalizing image intensities to range [0, 1]).

Furthermore, following [31], we augmented the data with the KiTS19 dataset [39] that contains high-quality CT kidney images of 210 subjects with their ground-truths. We manually separated the left and right kidneys in all KiTS19 dataset images, each of size 256 × 256 pixels. As such, the total numbers of training and validation images became 40,050 and 10,980, respectively. Both deep models were trained for 200 epochs using Adam optimizer starting out with a learning rate of 0.0001. This value was gradually reduced by a factor of 0.1 when the validation loss was not improved for 10 epochs. Furthermore, we used a dropout of 0.05 to avoid overfitting. Training was run on a workstation having two Nvidia GPUs along with two Intel Xeon Silver 4114 2.20 GHz CPUs and 128 GB of RAM. We then investigated the performance of the trained networks on the test subjects’ datasets. Table 2 gives a comparison between the results of our approach and the UNet and BCD-UNet models, each using three dense blocks.

As evident from Table 2, the proposed approach performed better than both UNet-based models did. It yielded higher mean DC and IoU values along with lower mean 95HD values. In particular, in light of the 95HD metric, the proposed approach was more than four times more accurate than the UNet and BCD-UNet models were. Moreover, the lower standard deviations of our approach reflect its more stable performance compared to that of both models. 

## 5. Conclusions

Precise renal segmentation from DCE-MRI data is a prerequisite processing step in renal diagnosis pipelines for patients who have chronic kidney disease. However, exploiting information concerning the kidney’s shape in this step mandates a registration operation beforehand for relating the shape model coordinates to those of the image to be segmented. Imprecise alignment of the shape model induces errors in the segmentation results. In this respect, we have proposed a new dual-task variational approach jointly performing kidney segmentation and registration in an automatic manner. This approach is believed to present the following contributions:It can be considered as the first approach in the literature to achieve accurate kidney segmentation and registration at the same time.It embeds FCM clustering within an LSet method in one variational approach; the membership degrees of the image pixels are updated during the LSet evolution process considering pixels’ intensities directly as well as prior shape probabilities. This promotes our approach’s performance.It can automatically manipulate the misalignment between the kidney in the input image and the SP-model.Thanks to employing smeared-out Heaviside and Dirac delta functions in the LSet method, the approach is able to accurately segment the kidney from the image regardless of where the contour has been initialized.It embraces an efficient statistical Bayesian parameter estimation method for SP-model construction, which can better address the cases of unobserved kidney pixels in the images while building the model.

Experiments conducted on numerous DCE-MRI images obtained from 45 patients verified the high performance of the proposed approach. The approach was demonstrated to be robust against contour initialization without tuning its parameters. In comparison against various recent LSet-based methods as well as two UNet-based models, our new approach has shown better and more consistent performance. 

Our current research is directed towards improving the proposed approach. LSet contour evolution is guided by a partial differential equation involving several weighting parameters. These weighting parameters need proper settings. In our experiments, their values were experimentally chosen and then fixed throughout all conducted experiments without further tuning. We plan to investigate other weighting strategies to systematically find out proper values for these weights, like the scheme proposed in [43]. 

## Figures and Tables

**Figure 1 biomedicines-11-00006-f001:**
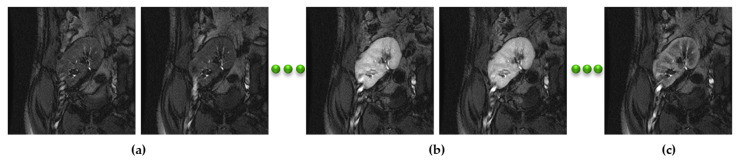
An example of DCE-MRIs of one subject depicts the contrast changes from the pre-contrast phase (**a**), moving on to the post-contrast phase (**b**), and ending with the late-contrast phase (**c**) during the perfusion of the contrast agent into the blood stream.

**Figure 2 biomedicines-11-00006-f002:**
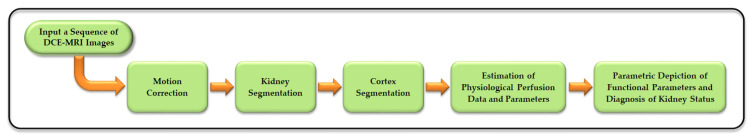
Block diagram of a typical DCE-MRI-based CAD system for the diagnosis of chronic kidney disease.

**Figure 3 biomedicines-11-00006-f003:**
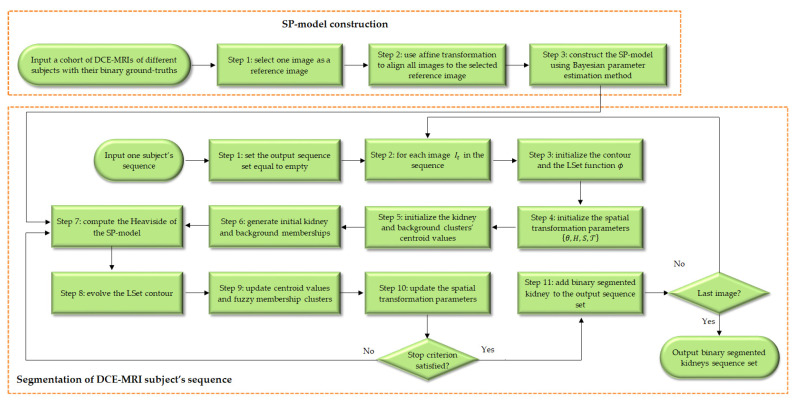
Flowchart of the proposed variational approach for DCE-MRI kidney segmentation and registration.

**Figure 4 biomedicines-11-00006-f004:**
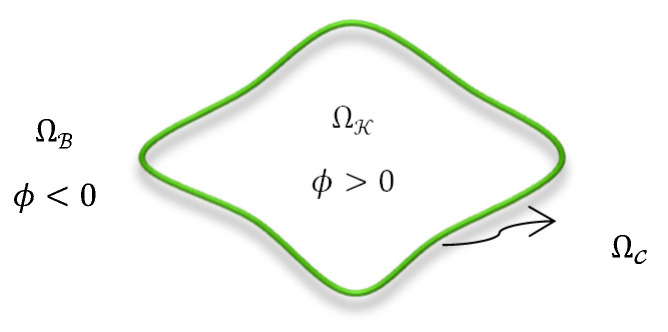
The representation of the Lset function in the image domain.

**Figure 5 biomedicines-11-00006-f005:**
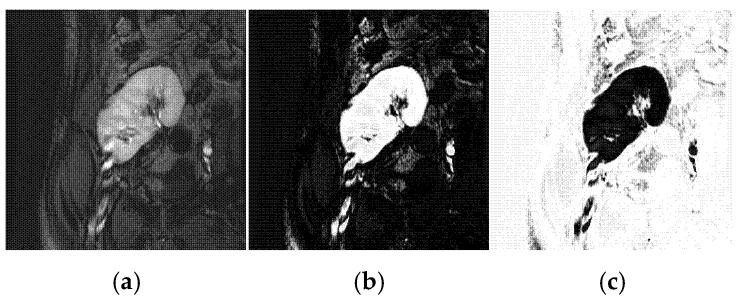
Segmentation by the FCMC algorithm of a DCE-MRI image. Column (**a**) depicts input kidney. Columns (**b**) and (**c**) show the output kidney and background clusters, respectively.

**Figure 6 biomedicines-11-00006-f006:**
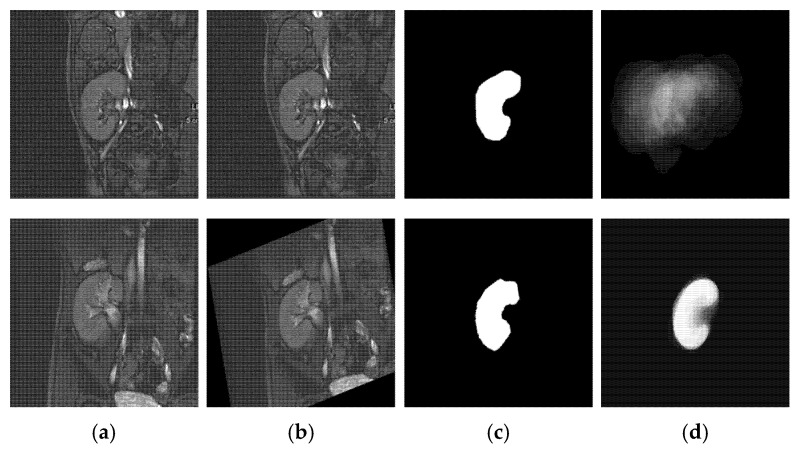
Illustration of the adopted strategy for SP-model construction: (**a**) sample training DCE-MRIs, (**b**) affine-registered DCE-MRIs, (**c**) manually segmented kidneys, and (**d**) the SP-model built from non-registered (**top**) and from affine-registered (**bottom**) DCE-MRIs.

**Figure 7 biomedicines-11-00006-f007:**
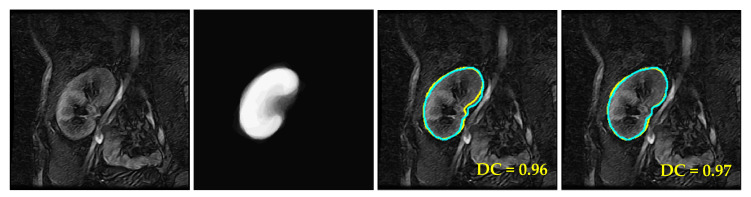
Example demonstrating the benefits of the joint segmentation and registration approach. (**a**) Two DCE-MRI input images, (**b**) constructed shape model, (**c**) segmented kidneys obtained by registration followed by segmentation, where the registration for the bottom image was not done accurately, and (**d**) segmented kidneys obtained by simultaneous segmentation and registration. The obtained kidney contours are marked in yellow, while the ground-truth segmentations are in cyan. The corresponding DC value is attached to each result.

**Figure 8 biomedicines-11-00006-f008:**
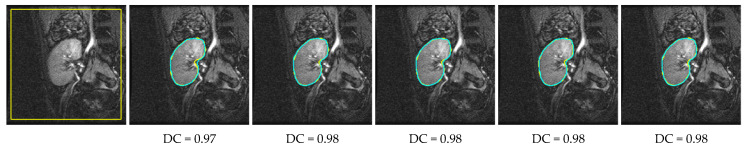
Output segmentations by the proposed approach and other existing LSet-based methods. (**a**) Input image with initial LSet contour. Results shown in yellow of the (**b**) FCMLS method [11], (**c**) PBPSFL method [12], (**d**) PSFL method [13], (**e**) FML method [17], and the (**f**) proposed approach. Ground-truth segmentation was imposed on each image in cyan along with the corresponding DC values.

**Figure 9 biomedicines-11-00006-f009:**
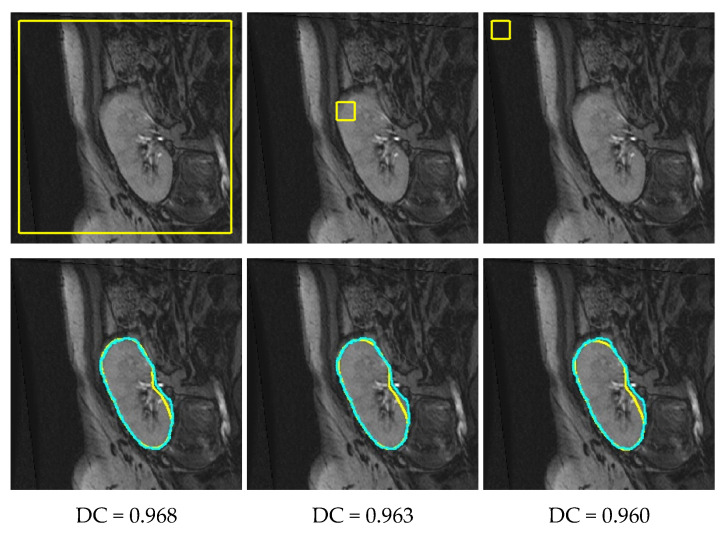
Example depicting how well the proposed approach performed with random initialized contours. Top row depicts a sample DCE-MRI scan with varying contour initializations in yellow, while the bottom row shows the output segmentations in yellow by the proposed approach. The ground-truth segmentations are overlaid in cyan, and the resultant DC value is given beneath each image.

**Table 1 biomedicines-11-00006-t001:** Performance comparison of the proposed approach and recent LSet-based methods.

Method	All DCE-MRIs			Affine-Transformed DCE-MRIs		
	DC	IoU	95HD	DC	IoU	95HD
FCMLS [11]	0.88 ± 0.10	0.79 ± 0.17	5.07 ± 7.65	0.83 ± 0.10	0.72 ± 0.14	8.35 ± 7.55
PBPSFL [12]	0.92 ± 0.06	0.87 ± 0.08	3.29 ± 5.65	0.90 ± 0.07	0.83 ± 0.09	5.4 ± 7.18
PSFL [13]	0.91 ± 0.06	0.84 ± 0.10	3.84 ± 4.56	0.87 ± 0.07	0.77 ± 0.11	6.57 ± 5.03
FML [17]	0.90 ± 0.08	0.83 ± 0.16	4.41 ± 6.4	0.87 ± 0.08	0.76 ± 0.12	7.3 ± 5.45
Proposed	**0.94 ± 0.03**	**0.89 ± 0.05**	**2.2 ± 2.32**	**0.93 ± 0.05**	**0.88 ± 0.06**	**2.5 ± 2.7**

**Table 2 biomedicines-11-00006-t002:** Performance of the proposed approach versus that of the UNet and BCD-UNet models.

Method	All DCE-MRIs			Affine-Transformed DCE-MRIs		
	DC	IoU	95HD	DC	IoU	95HD
UNet [26]	0.943 ± 0.04	0.90 ± 0.06	5.6 ± 15.7	0.946 ± 0.05	0.90 ± 0.05	3.5 ± 11.5
BCD-UNet [38]	0.942 ± 0.037	0.89 ± 0.06	4.4 ± 11.5	0.942 ± 0.035	0.89 ± 0.05	3.9 ± 10.3
Proposed	**0.95 ± 0.02**	**0.92 ± 0.03**	**1.0 ± 1.2**	**0.95 ± 0.025**	**0.91 ± 0.02**	**1.9 ± 2.1**

## Data Availability

Data are available upon request to the corresponding author.

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
