# Peer review of "Variational Approach for Joint Kidney Segmentation and Registration from DCE-MRI Using Fuzzy Clustering with Shape Priors"

_biomedicines, 2022, doi:10.3390/biomedicines11010006_

Round 1
Reviewer 1 Report
The paper proposes a new method for the segmentation and registration of kidney images using Fuzzy C-Means and LSet method.
The authors reported that the method proposed outperforms many state-of-the-art models including some based on UNets.
The paper in general is very interesting, well-written and seems to be correct.
However, a have a few comments to improve even further this paper:
- The introduction section needs to be well organized since it is slightly different from what we see in the literature. For example, Fig. 3 should be part of the results. Moreover, I recommend splitting this section in two where the second one would be "Related Works"
- Another suggestion is related to the tables with results (Table 1 and Table 2). Would be great if the authors highlighted the best results in each column. This makes it much easier to identify the best result.
Author Response
The authors like to thank the reviewer for the critical and constructive comments, which for sure have improved the content and presentation of the manuscript. Below we provide point-to-point responses to the comments and concerns. The relevant parts in the revised manuscript are highlighted in yellow.
1.1 Concern: The introduction section needs to be well organized since it is slightly different from what we see in the literature. For example, Fig. 3 should be part of the results. Moreover, I recommend splitting this section in two where the second one would be "Related Works".
Response:
Thanks for this suggestion. We have moved Figure 3 from the Introduction section and incorporated it into Subsection 4.2 of the Results section on Page 11 in the revised manuscript (Now it becomes Figure 7). Per the reviewer’s suggestion, we also split the introduction section into two parts. The second part now represents Section 2 on Pages 2-4 in the revised section, where we briefly review several DCE-MRI kidney segmentation methods developed in the literature. In addition, we discuss the research gap and the paper contributions in this section.
1.2 Concern: Another suggestion is related to the tables with results (Table 1 and Table 2). Would be great if the authors highlighted the best results in each column. This makes it much easier to identify the best result.
Response:
Thanks a lot for this suggestion. We have highlighted the best results in Tables 1-2 in the revised manuscript in bold.
Reviewer 2 Report
In this paper, a variational approach is proposed for joint kidney segmentation and registration, which is based on fuzzy clustering with shape priors. Experiments have been conducted to show the effectiveness of the proposed method. Some comments can be found as follows:
1. The main contributions can be summarized at the end of the introduction part, instead of the conclusion.
2. It would be better to have a figure to illustrate the whole framework of the proposed model.
3. Figure 4 does not been mentioned in the texts.
4. It is recommended to review some other medical image processing tasks, such as optimum weighted multimodal medical image fusion using particle swarm optimization (image fusion), RTN: reinforced transformer network for coronary CT angiography vessel-level image (image quality assessment), etc.
5. The organization of this paper would be further improved. For example, the adopted data can be described after the method statements, which is at the beginning of experiments.
Author Response
The authors like to thank the reviewer for the critical and constructive comments, which for sure have improved the content and presentation of the manuscript. Below we provide point-to-point responses to the comments and concerns. The relevant parts in the revised manuscript are highlighted in yellow.
2.1 Concern: The main contributions can be summarized at the end of the introduction part, instead of the conclusion.
Response:
Thank you for this suggestion. We have highlighted the main contributions of the proposed approach at the end of Section 2 on Pages 3-4.
2.2 Concern: It would be better to have a figure to illustrate the whole framework of the proposed model.
Response:
Thanks for this suggestion. We have added Figure 6 at the end of Section 3 on Page 10 in the revised manuscript. This figure shows the flowchart of the proposed approach.
2.3 Concern: Figure 4 does not been mentioned in the texts.
Response:
We mentioned this figure in Subsection 3.2 on the bottom of Page 4 in our manuscript.
Note that we have renumbered all figures in the revised manuscript so that this figure becomes now Figure 3.
2.4 Concern: It is recommended to review some other medical image processing tasks, such as optimum weighted multimodal medical image fusion using particle swarm optimization (image fusion), RTN: reinforced transformer network for coronary CT angiography vessel-level image (image quality assessment), etc.
Response:
Thanks a lot for referring us to these references. We do agree. As almost all the existing level set-based methods, our new approach includes some weighting parameters to reflect the contributions of the different components in the partial differential equation governing the level set evolution. The values of these parameters are experimentally chosen after performing several preliminary experiments, and then are not changed or further tuned throughout all conducted series of experiments. However, as the reviewer suggested, we can use other weighting strategies to systematically find out proper values for these weights, like the one proposed in the first-mentioned paper by Shehanaz et al in Optik.
We note this issue in the Conclusions Section on Page 15 in the revised paper in the context of discussing the proposed method’s limitations and how to overcome them in our ongoing and future research efforts. The above reference by Shehanaz et has been added to our reference list.
2.5 Concern: The organization of this paper would be further improved. For example, the adopted data can be described after the method statements, which is at the beginning of experiments.
Response:
Thanks a lot for this suggestion. We have re-structured the revised manuscript so that the data used in our study is now described in Subsection 4.1 in the Results section on Page 10.
Round 2
Reviewer 2 Report
The authors have addressed most of my comments. One minor point is that it is better to state the whole proposed framework and then explain each steps.
Author Response
The authors like to thank the reviewer for the critical and constructive comments, which for sure have improved the content and presentation of the manuscript. Below we provide point-to-point responses to the reviewer's comments and concerns. The relevant parts in the revised manuscript are highlighted in yellow.
1.1 Concern: One minor point is that it is better to state the whole proposed framework and then explain each step.
Response:
Thanks a lot for this suggestion. Per the reviewer’s suggestion, we moved Figure 6 which illustrates the flowchart of the proposed approach from the end of Section 3 to Section 3.2 on Pages 4-5, where we begin describing our proposed approach. As such, the flowchart figure now becomes Figure 3. We give an overview on our approach in Section 3.2, while we detail all the steps of the approach in the subsequent subsections (Section 3.3 through Section 3.6).